# Diet for Human and Planetary Health: Why We Should Consider Limiting Meat?

**DOI:** 10.3390/ijerph22101499

**Published:** 2025-09-29

**Authors:** Hamsika Moparty, Manya Pala, Sahaja Ampolu, Swapna Gayam

**Affiliations:** 1Division of Liver Disease & Transplant Hepatology, Rutgers New Jersey Medical School, Newark, NJ 07103, USA; hamsika.moparty@gmail.com; 2Department of Internal Medicine, Southern Regional Medical Center, Riverdale, GA 30274, USA; manyapala@gmail.com; 3West Virginia University School of Medicine, Morgantown, WV 26506, USA; sa00138@mix.wvu.edu; 4Section of Gastroenterology and Hepatology, West Virginia University, Morgantown, WV 26506, USA

**Keywords:** meat-based diet, plant-based diets, human health, sustainability, biodiversity

## Abstract

Climate change is currently the most significant threat to public health, and human activities are the major contributing factor. There is an urgent need to prioritize mitigation strategies at both personal and public policy levels. There is a general lack of belief that changes at a personal level would have a significant effect. However, it is vital to recognize the importance of food consumption on one’s personal footprint and how it can be used as a key feature in mitigation efforts. The Center for Sustainable Systems at the University of Michigan projects that reducing meat consumption per individual by 50% reduces an individual’s carbon footprint by 35% per day and reducing by 90% cuts an individual’s carbon footprint by 51% per day. Additionally, high meat consumption has been associated with increased risks of cardiovascular disease, diabetes, and colorectal cancer. In contrast, plant-based diets are linked to better health outcomes and lower mortality rates. This article is a narrative review and reviews current evidence on the health and environmental impacts of meat-based diets and highlights the potential benefits of plant-forward dietary patterns. These findings support the integration of dietary recommendations into climate and public health strategies. Promoting plant-based diets through clinical guidance and policy initiatives may offer a cost-effective, scalable approach to advancing both population health and environmental sustainability.

## 1. Introduction

Diet plays a key role in both human and planetary health. Modern meat-heavy diets have been shown to be detrimental to human health, especially processed meats, which are defined as products usually made of red meat that is cured/salted or smoked and often contains high quantities of minced fatty tissue [1]. Anthropogenic activities related to meat production have accelerated the emission of greenhouse gases (GHGs) resulting in debilitating effects on planetary health. This review discusses the effects of meat production and consumption on the health of people and the planet compared to plant-based diets. The goal is to highlight the potential of dietary change as a strategy to improve public health and support climate mitigation.

## 2. Methods

This is a narrative review with some elements of a scoping review included.

An extensive search was conducted for articles on PubMed, Google Scholar, Science Direct and WVU Library databases between June 2021 and July 2025. Approximately 100 studies, magazine articles and online reports published between 1999 and 2023 were reviewed, out of which 55 were found to be relevant and chosen for the purpose of this manuscript.

Inclusion criteria for the search: Studies related to the effect of meat-heavy diets on human health; studies on the effect of plant-based diets on human health; plant-based alternatives to meat proteins; effect of animal farming on the environment; effect of meat processing on the environment; climate mitigation strategies for animal farming and processing.

Exclusion criteria for the search: Studies related to non-dietary causes of human cardiovascular, gastrointestinal diseases and cancer; studies related to human activities other than meat production and processing associated with climate change; studies describing climate change mitigation strategies not related to animal farming or processing; studies related to non-dietary strategies to improve human health.

Key words: climate change; meat-based diet; plant-based diets; omnivarian diet; carcinogens; human health; planetary health; greenhouse gases; environmental impact; livestock farming; climate mitigation; sustainability; flexitarian diet; sustainable diets; biodiversity; climate mitigation.

## 3. Discussion

### 3.1. Effect of Plant and Meat-Based Diets on Human Health

#### Effect of Plant-Based Diets on Human Health

Research has shown that plant-based diets lower the risk of obesity, diabetes and cardiovascular diseases (CVDs) like coronary artery disease and cerebrovascular accidents by 20–24% [2,3,4,5,6,7]. Reducing the incidence of high-risk conditions like obesity may even help reverse diabetes and advanced cardiovascular diseases [8,9,10,11,12]. A recent Danish prospective cohort study concluded that dietary inorganic nitrates in green leafy vegetables and beets are excellent sources of nitric oxide (NO), associated with lower baseline systolic and diastolic blood pressure and reduced risk of CVD including peripheral arterial disease (PAD), ischemic heart disease, ischemic stroke and heart failure due to the vascular effects of inorganic nitrate [13]. Studies have shown that plant-based diets are inversely related to the risk of type 2 diabetes and coronary heart disease [14,15]. In one of the largest studies performed on diets, vegan, vegetarian, pescatarian and semi-vegetarian diets were shown to have a 12% lower overall mortality risk compared to omnivarian diets [15]. A recent twin study by Stanford researchers on the effect of healthy vegan vs. healthy omnivarian diets on 22 sets of twins showed that twins randomized to the vegan diet had a significant reduction in low-density lipoprotein cholesterol concentration (LDL), fasting insulin level and body weight in 8 weeks [16].

### 3.2. Effects of Meat-Based Diets on Human Health

#### 3.2.1. Cardiovascular Health

Conversely, meat-based diets increase the risk of CVD. Meats contain environmental pollutants like polychlorinated dibenzo-p-dioxins and dibenzofurans (PCDD/F), dioxin-like polychlorinated biphenyls (PCB) and polychlorinated naphthalenes (PCN), which are highly soluble in fats and increase the risk of CVD and cancers [1,17,18]. Micha et al. noted that processed meats contain approximately 400% more sodium and 50% more nitrates per gram. Nitrates and peroxy-nitrites promote endothelial dysfunction, atherosclerosis and insulin resistance, which are precursors of diabetes. The study noted a 51% increased risk of diabetes mellitus (DM) with a 50 g increase in daily processed meat consumption. Further, processed meats contain high amounts of saturated fat and cholesterol, thereby increasing risk of hypertension, cardiovascular diseases and cerebrovascular accidents (CVA) [19].

#### 3.2.2. Gastrointestinal Health

With regard to meat consumption and gastrointestinal disorders, Chan et al. published several studies with strong evidence that diets rich in red meats increase the incidence of diverticulitis [20]. A recent large prospective cohort study on more than 46,000 male health professionals by this group concluded that a diet rich in vegetables and fiber and low in red and processed meats has overall anti-inflammatory effects and may help reduce the risk of diverticulitis [21].

Association between red meat consumption and ulcerative colitis (UC) has been noted in multiple studies. Colonic metabolism of meat heme and amino acids into toxins like hydrogen sulfide, etc., and an unhealthy change in gut microbiome (colonic dysbiosis) secondary to meat consumption have been implicated as risk factors for UC [22,23]. It is now well known that a healthy gut microbiome is essential for overall human health and immunity down to the genetic level [24]. It is important not only for maintenance of proper structure and function of the gastrointestinal tract, but also for the development of gut mucosa and synthesis of certain vitamins. Dysbiosis of gut microbiome has been associated with a range of gastrointestinal diseases like inflammatory bowel disease (IBD), irritable bowel syndrome (IBS) and metabolic diseases like obesity and diabetes. Diet is the most essential factor in the development of gut microbiome and diets rich in fruits and vegetables are critical for a rich, diverse and healthy microbiome due to prebiotic effects. In contrast, meat-based diets diminish the health and diversity of the human microbiome [25]. Gut microbiome can also affect the development of colorectal cancer by integrating environmental factors with host physiology, immune and metabolic signal changes [25]. Song and Chan showed how diets rich in red and processed meats and low in fiber (Western diet) may influence carcinogenesis in colorectal cancer through gut microbiota-related mechanisms [26].

A possible association between processed meat and IBS has been studied with IBS subjects consuming significantly more canned food (*p* < 0.01) and processed meat (*p* < 0.01) [27].

Eliminating mammalian meat consumption would also benefit patients with alpha-gal syndrome, which has gained attention in recent years. It is an IgE-mediated delayed hypersensitivity to galactose sugar in mammalian meat, secondary to a lone star tick bite, manifesting with skin and gastrointestinal (GI) symptoms. Ironically, the origins of this syndrome could also be traced back to climate change causing expanding tick seasons, geographically and temporally [28].

#### 3.2.3. Association of Meat-Based Diets and Cancer

Research has shown an association between meat consumption and cancer. Veettil et al. conducted an umbrella review and found an association between colorectal cancer (CRC) and higher meat consumption, and inverse association with fiber-rich diets like the Mediterranean diet [29]. Consuming approximately 50 g of processed meat per day would increase CRC risk by 18% [30]. In 2015, processed meats were classified as a Group A carcinogen (carcinogenic to humans) and red meat as a Group 2B carcinogen (probably carcinogenic to humans) with respect to CRC by the International Agency for Research on Cancer (IARC) [30]. Newer studies have also shown a link between processed and red meat and breast/pancreatic/prostate/esophageal/gastric cancers [31]. Ma et al. found red meat was associated with an 84% increased risk of hepatocellular cancer in two prospective cohorts [32]. Total, red, and processed meat intake has been associated with an increased risk of gastric adenocarcinoma (GAC) in several studies, and potential underlying biological mechanisms have been identified for this association, but evidence is still limited. Moreover, meat cooking practices and doneness preferences, which have been less studied, could independently increase the risk of GAC and might help in explaining the heterogeneity currently observed among results of epidemiological studies [33].

The American Institute for Cancer Research states that more than 18 oz of red meat consumption per week can increase cancer risk and recommends skipping processed meats [34]. The American Cancer Society therefore recommends ‘a healthy eating pattern for all’ including a variety of vegetables and avoiding red and processed meats, to prevent numerous cancers [35]. According to dietary experts at Johns Hopkins University, healthy ways to prepare vegetables in order to retain the most nutrients are steaming, pressure cooking, microwaving and stir-frying [36].

There is a paucity of studies looking at the mechanistic processes involved in carcinogenesis associated with meat consumption. Implicated compounds and proposed pathogenic mechanisms are summarized in Table 1 [1,31,37].

### 3.3. Impact of Food Production on the Ecosystem

#### 3.3.1. Overall Production of Greenhouse Gas (GHG) Emissions and Global Warming

Global food production significantly contributes to GHG emissions and climate change by altering natural biodiversity, consuming fresh water, influencing land microbiome with chemical and antibiotic use, and altering nitrogen and phosphorus cycles. According to EAT-Lancet Commission’s Report in 2019, “vegan and vegetarian diets are associated with greatest reductions in GHG compared to animal-based diets such as beef, lamb and mutton”. This commission recommends increasing consumption of plant-based foods and reducing consumption of meat for the health of people and the planet. Overall, plant-based foods have lower negative environmental impacts per unit weight, per serving, per unit of energy, or per unit protein weight compared to all animal-derived foods [38].

Clune et al. expanded on this topic by performing life cycle assessment (LCA: in-depth methodology for assessing environmental impact) of various foods. Their study showed that grains, fruits and vegetables have the lowest environmental impact per serving. Meat from ruminant livestock (cattle, sheep, goats) has the highest environmental impact [39]. Poore and Nemecek calculated GHG emissions per kilogram of food product and not surprisingly, beef-herd-beef tops the list at 60 kg CO_2_ followed by lamb/mutton at 24 kg CO_2_e and dairy-herd-beef at 21 kg CO_2_ [40]. A meat-heavy diet emits approximately 7.19 kg CO_2_e/day while a vegetarian diet emits almost half, about 3.81 kg CO2/day [41]. Sabate et al. calculated the environmental costs associated with 1 kg protein from beef compared with 1 kg of kidney beans. It was found that 1 kg of beef requires 18 times more land, 10 times more water, 9 times more fuel, 12 times more fertilizer and 10 times more pesticides than the same amount of protein from kidney beans [42]. A Stanford report in 2021 noted that a single quarter pound hamburger patty requires approximately 460 gallons of water to produce. This report also notes that the environmental impact of everyone in the United States giving up meat and cheese once a week is equal to taking 7.6 million cars off the road [43].

Globally, Oxford researchers calculated that by 2050, approximately 8 million lives, upwards of USD 1.5 trillion from climate-related damages and 60–70% food-related emissions would be saved by adopting vegetarian and vegan diets [44]. Despite these facts, global meat consumption is projected to reach up to 500 million tons by 2050 [45]. The impact of global food system emissions alone would make it impossible to limit global warming even if we eliminated fossil fuels right away according to Clark et al. [46].

#### 3.3.2. Effect of Livestock Production

Livestock production is one of the highest contributors to GHG emissions and global warming, approximately 18% [47]. There are many ways that global meat consumption adversely affects the environment (listed below).

First, livestock farming leads to deforestation [47,48,49]. Huge areas of forests are being cleared worldwide for livestock farming, both for pasture lands and to produce feed crops. Livestock produces 18% of the world’s calories and 37% of the world’s protein, yet accounts for 77% of the world’s agricultural land [48]. Rather than the Amazon rainforest continuing to serve as a carbon sink, deforestation for livestock farming is leading to significant carbon emissions.

Second, mammals produce CO_2_ through exhalation, methane via flatulence, and nitrous oxide and ammonia via urine and manure. In fact, enteric emissions make up 70 and 75% of beef and lamb GHG emissions, respectively [46,47,50]. According to the Food and Agricultural Organization of the United Nations (FAO) report on ‘Livestock’s role on climate change and air pollution’, the livestock industry produces 35–40% of world’s methane and 65% of nitrous oxide [47]. Lactating cows are the highest livestock producers of methane [51]. The significance of this fact is that methane and nitrous oxide (N_2_O) have significantly more global warming potential than CO_2_, 25 and 298 times, respectively [47].

Third, when large amounts of animal waste accumulate, the resulting ammonia breeds bacteria. These bacteria combine with other pollutants like N_2_O, sulphur dioxide, and chlorofluorocarbons to form nitric acid and sulphuric acid resulting in acid rain. Acid rain destroys soil, crops, forests and water life [52].

Fourth, antibiotics that are used in raising livestock also disperse into the atmosphere along with antibiotic resistance genes (ARGs) and contribute to antibiotic resistance. This has been shown by researchers at Texas Tech University among others [53].

Fifth, meat-processing plants use energy, water, chemicals and packaging like all food-processing plants, but the difference is that meat processing requires massive amounts of water and energy, given that meat is highly perishable and needs to stay cool and clean. Meat processing also leads to large amounts of solid organic waste and wastewater with high biochemical oxygen demand leading to pollution of soil, air and water, depletion of aquatic biodiversity and eutrophication [47,48].

Sixth, livestock is responsible for freshwater usage, i.e., more than 8% of global water, mostly for feed crops. Meat and dairy industries alone use one-third of global fresh water. It is a major contributor to water pollution from animal waste, antibiotics, chemicals and pesticides used for feed crops. Finally, fossil fuels used in feed and livestock product manufacture, processing, transport and marketing also indirectly cause carbon release [47]. In total, the livestock industry is responsible for a staggering 18% of GHGs generated by human activities worldwide. This is more than all the cars, trucks, airplanes and ships in the world combined [47].

Livestock farming also negatively impacts biodiversity. The price of thriving human civilization is paid by dwindling populations of wild mammals, birds, fish and trees, significantly altering earth’s ecological balance [54]. In 2020–2021, 96% of the world’s mammal biomass (excluding humans) was livestock and human-preferred animals [49]. According to the International Union for Conservation of Nature (IUCN) Red List of Threatened Species, out of the 28,000 threatened species, 24,000 are listed as threatened by agriculture and aquaculture [49]. In general, livestock farming has decimated global wild mammal population.

In addition to the massive global environmental impact, livestock farming has significant impacts at the local level, affecting humans who live in and around these farms. Roughly 80% of ammonia emissions in the US can be linked to animal waste. In the US, most large-scale livestock farming happens in huge ‘factory farms’ where hundreds and thousands of cattle are kept together, which poses not only air quality threats but miserable living conditions to local communities due to the unpleasant smell of noxious gases. The health effects of livestock farming on farm workers and local residents are numerous and include asthma, bronchitis, cardiac effects, and zoonotic infections [50]. A recent study published by Domingo et al. noted that nearly 13,000 deaths annually in the US can be attributed directly or indirectly to food-related fine particulate matter (PM_2.5_) generated by meat production. They note that shifting to a predominant plant-based diet can reduce agricultural air quality related deaths by 68–83% [55].

## 4. Potential Solutions

The obvious solution of reducing meat consumption is a daunting task. Beef, lamb and goat are among the top five meats consumed in the world.

### 4.1. Policy

Similar to other public health initiatives, health care providers should seek active involvement in health care policy and lead the effort on national and global scales. Hospitals and healthcare facilities should increase plant-based items and limit carcinogenic meats on their menu. Akin to anti-smoking messages, hospitals could broadcast that their meals are served with plant-based proteins to promote health. This would promote both patient health and patient awareness. Further, health care providers should also recommend reduced meat consumption and encourage plant-based diets to patients.

### 4.2. Health

Meat has traditionally been the dominant source of dietary protein in Western countries. This is probably based on the myth that meat is the best protein source. Lack of awareness about the negative effects of meat consumption and about the availability of alternative and healthier protein sources is probably one of the most important contributing factors. Reassuring patients about the availability of nutritious and widely available non-animal sources of protein is a start. Beans, lentils, chickpeas, tofu, tempeh, edamame, nuts, grains and many more foods are excellent sources of plant-based protein. All plant foods contain the nine essential amino acids (histidine, lysine, leucine, isoleucine, methionine, phenylalanine, threonine, tryptophane, valine) required to make proteins and many of them like soy, nuts, beans, seeds and non-dairy milk products contain protein amounts similar to animal sources [41]. Lentils and beans contain fiber that helps in weight management and blood sugar regulation that can help reduce hypertension, diabetes and cardiovascular disease. This is in addition to prebiotic, laxative and overall gut benefits [56]. Plant-based “meats” or meat alternatives can help with the transition away from meat-heavy diets.

While there are several health benefits of switching to plant-based diets, such a switch may have some pitfalls: patients with irritable bowel syndrome may experience functional symptoms including abdominal pain, bloating and changes in bowel habits secondary to the sudden increase in fiber in diets and possible food malabsorption of increased sugars like fructose. Making small and gradual changes could help overcome some of these side effects [57].

### 4.3. Environment

Plant-based food production creates a lower carbon footprint as mentioned earlier. However, eating less meat would mean a major lifestyle change for many. Transitioning to meat from non-ruminants like poultry (6 kg CO_2_) and fish (farmed fish 5 kg CO_2_, wild caught fish 3 kg CO_2_), which have a much lower carbon footprint per kg of food product, could also be considered as an intermediate step [40]. This allows for a more adjustable lifestyle and not completely eliminating meat from one’s diet. A ‘flexitarian’ lifestyle with flexible eating habits is a more reasonable goal for many heavy-meat eaters.

## 5. Conclusions

Climate change and health are closely connected. Food consumption is a vital component of one’s personal footprint and it can be used as a key feature in mitigation efforts. Reducing meat intake and promoting plant-based diets can improve health outcomes and reduce the environmental damage caused by food production. This narrative review consolidates numerous studies that provide evidence-based research regarding the negative health and environmental effects of meat-heavy diets, in an effort to emphasize the urgent need to include dietary change in both public health and climate policy for a more sustainable future. Public health research on sustainable farming and dietary recommendations could help guide government policy on human health and environmental protection.

## Figures and Tables

**Table 1 ijerph-22-01499-t001:** Proposed carcinogenic compounds and mechanisms induced by meat consumption.

Methods of Processing Meat	Compounds in Unprocessed and Processed Meat	Proposed Mechanisms Involved in Carcinogenesis
Salting	HCBs	Salting → NOCs → interaction with gastric carcinogens (e.g., HP)
Curing	PCBs	N-nitrosation → oxidative load and lipid peroxidation → DNA adducts
Smoking	PCDD/Fs	Heme iron → nitrosylation → NOCs
High-temperature cooking (smoking, grilling, barbecuing)	Heme iron	Heme iron → oxidative stress → lipid peroxidation → protein modification →DNA damage →DNA adducts
	Mercury	
	Arsenic	PAHs → radical cation pathway/diol-epoxide pathway/ortho-quinone pathway → radical cations, diol-epoxides, ortho-quinones → DNA, RNA, glutathione adducts → DNA mutations, alterations in gene expression, carcinogenesis
	Lead	
	NOCs	
	PAHs	
	HAAs	

HCBs = hexachlorobenzes, PCBs = polychlorinated biphenyls, PCDD/F = polychlorinated dibenzo-p-dioxin and dibenzofuran, NOCs = N-Nitroso-compounds, PAHs = polycyclic aromatic hydrocarbons, HAAs = heterocyclic aromatic amines, HP = Helicobacter pylori; DNA = deoxyribonucleic acid.

## Data Availability

Not applicable.

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
