# Peer review of "Diet for Human and Planetary Health: Why We Should Consider Limiting Meat?"

_ijerph, 2025, doi:10.3390/ijerph22101499_

Round 1
Reviewer 1 Report
Comments and Suggestions for Authors
This is a well-timed and important review that addresses the dual health and environmental implications of meat-based diets. The manuscript is supported by strong literature and policy-relevant framing, and it integrates a rich variety of evidence from health and sustainability research. However, the scientific contribution would be improved by a clearer articulation of the methodology used for selecting and assessing sources. At present, the review lacks a structured search strategy or quality assessment criteria, which weakens transparency and reproducibility.
I recommend the following minor revisions:
-
Strengthen the methodology section: clarify the timeframe of the search, inclusion/exclusion criteria, and number of studies reviewed.
-
Adjust the tone in several parts to be more academic and less advocacy-driven (e.g., avoid informal phrases such as “Eat less meat is the most obvious answer”).
-
Reorganize Section 3 into clearer sub-sections for better readability (e.g., split into Health, Environment, Policy).
-
Shorten and synthesize the Conclusion to avoid redundancy.
-
Expand the list of keywords to better reflect the breadth of the article (e.g., add “climate mitigation,” “flexitarian,” “sustainable diets”).
The manuscript has great potential to inform both public health recommendations and environmental policy, and with some polishing, it will meet the standards of IJERPH.
Comments on the Quality of English LanguageThe manuscript is generally well written and understandable, but the quality of English can be improved for clarity and academic tone. There are several instances of informal or conversational phrasing (e.g., “Eat less meat is the most obvious answer”, “horrid smell”) that should be revised to maintain a consistent scientific style. Sentence structure can also be refined in certain sections to enhance readability and precision. A professional language editing pass is recommended to ensure fluency, eliminate redundancy, and elevate the academic rigor of the text.
Reviewer 2 Report
Comments and Suggestions for Authors
The article is written on a relevant topic. The tasks are appropriate. The methods have been selected accordingly to reveal the topic of the work, accomplish the tasks, and achieve the goal. The author set the task of studying the importance of food consumption for personal impact and how it can be used as a key element in mitigation efforts. On this basis, data from the University of Michigan’s Center for Sustainable Systems were used. The article predicts that reducing meat consumption per person by 50% decreases the individual carbon footprint by 35% per day, while a 90% reduction decreases the individual carbon footprint by 51% per day. The research results in the article allowed the authors to state that high meat consumption is associated with an increased risk of cardiovascular diseases, diabetes, and colorectal cancer. In contrast, plant-based diets are linked to improved health status and reduced mortality rates. The authors paid sufficient attention to recent data on the impact of meat-based diets on health and the environment, as well as highlighted the potential benefits of dietary patterns focusing on plant-based foods. The conclusions drawn confirmed the hypothesis regarding the integration of dietary recommendations into climate and health strategies. Promoting plant-based diets through clinical guidelines and policy initiatives can offer a cost-effective and scalable approach to improving both population health and environmental sustainability.
However, the article requires revisions:
- The article includes only one table — Table 1. Proposed carcinogenic compounds and mechanisms induced by meat consumption, which describes carcinogenic compounds and mechanisms caused by meat consumption. However, there is a lack of analysis regarding the use of plant-based diets.
- It is advisable to add economic-mathematical models to the article that, based on data analysis, would confirm the authors’ conclusions regarding “a 50% reduction in meat consumption per person decreases the individual carbon footprint by 35% per day,” “a 90% reduction decreases the individual carbon footprint by 51% per day,” “high meat consumption is associated with an increased risk of cardiovascular diseases, diabetes, and colorectal cancer,” “plant-based diets are linked to improved health and reduced mortality,” and “the impact of meat-based diets on health and the environment.” The authors make these assumptions, but digital confirmation is lacking.
- The “Conclusions” section should be expanded to emphasize previous authors’ research and how the authors of the article advance previous studies in this area or, conversely, refute theses from earlier research.
The manuscript is clear, relevant to the field, and well-structured. It lacks a sufficiently substantiated mathematical basis, which is likely absent.
The citations are current.
The conclusions do not fully correspond to the presented evidence and arguments and require revision.
Reviewer 3 Report
Comments and Suggestions for Authors
Dear Authors
Thank you for your effort on the topic of an environmentally-friendly diet. Overall, I think your review is written well, but it lacks an innovative view on the problem. It can be changed, although. I would recommend you focus on the possible plant-based diet (or semi-plant-based diet) implementation methods in public health, rather than repeating already known statements about reducing meat production and consumption advantages. Unfortunately, the article in this form does not present new facts or new ideas, and, in its current form, it does not present possible changes for the healthcare system or environmental health improvement. I would also recommend you polish the description of the methods used to prepare your manuscript. This section should be more detailed. Please add information about the entire searching process (even if you prepared a narrative review), the range of years when articles were published, how many articles you used, the criteria of exclusion and inclusion, and for how long you were working on it (these that were applicable in your work). Please also write the type of review the Authors prepared. I would also suggest using more formal, professional language, for example, in line 256 or 248 (replace “horrid” with another word).
I additionally noted below additional comments about precise lines and/or sections:
Lines 32-34: Sentences “The health benefits of plant-based diets are well known. On the other hand, modern meat-heavy diets are causing a deterioration of human health.” These two statements have the same message: a plant-based diet gives benefits to human health compared to a meat diet. I do not think “on the other hand” is used well between these two sentences.
Lines 50-53: These two points are the conclusion of your work. Writing them separately in this section suggests that you analyzed possible guides for public health and governmental policies. If you do it, you can leave these points written here then. Otherwise, I suggest removing them to the paragraph with the conclusions.
Line 54: Methods: This section should be improved as I have already mentioned above. You have to explain your methodological thinking, even if you did not prepare a systematic review.
Lines 60 and 61: I recommend using one way of writing a diet: “plant based diet” or “plant-based diet”.
Lines 60-64: The information about the risk of obesity and cardiovascular diseases is repeated. I suggest improving the stylistic of this part or deleting the first sentence.
Line 67: The abbreviation “CVD” should be used in the first place when “cardiovascular disease(s)” is used, and the abbreviation can be written then. Please correct. Similar comments are for: lines 104, 112, 118,
Lines 64-69: You mentioned Danish research here about the impact of eating green leafy plants on health indicators. It would be better to write here exactly what vegetable species or what forms of them were used in the dietary intervention, instead of writing the proverb.
Lines 59-76: The entire paragraph 3.1. is quite poor in information while 3.2. is detailed and long. Please correct it to keep a similar length and capacity of content in 3.1. and 3.2. Other way to improve this section is to merge it with 3.2. You can treat this section as a review of the effects of both plant and animal foods on human health.
Line 86: “DM” should be written as a full name with the abbreviation in brackets because it is the first place where you use it.
Line 96: “red meat” instead of “read meat”. Please correct.
Lines 97-99: I suggest improving the statements referred from positions 22 and 23. Overall mentioned groups such as hydrogen sulfide, phenolic compounds and amines can have a good impact on human health as well. Please specify which compounds from these groups had toxic effects.
Lines: 105-16: “(…) diets rich in fruits and vegetables are critical for a rich, diverse and healthy microbiome”. That is true mainly because of the high content of fibre in vegetables and fruits. It would be worth explaining here why vegetables and fruits make the gut microbiome richer.
Lines 113-114: Except for the p-value, it is not understandable what numbers in brackets mean. This information can be worth. It is because you wrote here exactly measurable effects. However, you have to explain the result from the article referred here. Otherwise, it is unusable.
Line 126: The definition of processed meat / high-processed meat (or what you mean by “processed meat” in this review) should be written in the introduction of the review. Please correct it. If reviewed articles had other special definition of such products, you can write exactly name of this products in proper places, for example sausages, hams, canned meat, salty meats, mixed minced meat etc.
Line 142: Small error in reference.
Lines 142-144: You reviewed the eating of meat prepared in different cooking ways. It is obvious that culinary treatment can change the quality of meat differently (e.g., because of different temperatures). In this line, when you mentioned ACS and their healthy eating pattern, you can also write what culinary treatment of vegetables is recommended by professionals. This change can make this section more complex and informative.
Entire section 3.2.: In this section the information about polycyclic aromatic hydrocarbons is missing. These compounds have carcinogenic properties, and they are common in meat prepared at very high temperatures (such as on a barbecue, grill, or frypan). I saw that HAA or PAH are noted in Table 1. However, this table seems to be unfinished.
Lines 152-153: In my opinion, this statement is brachylogy. It should be proofed by adding the reference in the end of this sentence. If it presents Authors’ thoughts, it should be clearly written here. Otherwise, please cite here the results from the article(s) that present an authoritative reduction of strain on healthcare systems.
Line 154: What kind of environmental savings do you mean? The impact of food on the environment is presented later. In my opinion, it is not necessary to make here the relation between this section and 3.3. If you prefer keeping this line up here, I recommend you explain how the facilitation of the healthcare system can be a direct addition to the environment.
Lines 158-156: In the case you have one citation for a subparagraph, you do not have to refer to position 37 twice.
Lines 176-177: It would be better to split this long sentence for two, after the “(…) Sabate et al.”.
Lines 156 and 157: Titles are not understandable. I recommend changing them. The exact recommendation is noted below as the comment for “The entire paragraph 3.3.”.
Line 237: “human-preferred”, instead of “human preferred”.
Line 252: The number next to the “PM” is not understandable. Please correct it.
The entire paragraph 3.3.: In my opinion, the names of sections can be improved. I would suggest rewriting them: “3.3. Impact of food production on the ecosystem”, “3.3.1 Overall Production of greenhouse gas (GHG) emission and global warming” (it will contain lines 156 -189), “3.3.2. Effect of livestock production” (it will contain lines 190-254). Merging these sections is highly recommended to make your text more structured.
Lines 269-274: It is written too generally. I would recommend writing down in the text which amino acids exactly exist in which raw plant food materials. What other nutritional benefits are possible to obtain from lentils or beans? Do they have strong benefits for the environment compared to meat products?
